# Myosin motors fragment and compact membrane-bound actin filaments

**Sven K Vogel\*, Zdenek Petrasek, Fabian Heinemann, Petra Schwille\***

Max Planck Institute of Biochemistry, Department of Cellular and Molecular Biophysics, Martinsried, Germany

**Abstract** Cell cortex remodeling during cell division is a result of myofilament-driven contractility of the cortical membrane-bound actin meshwork. Little is known about the interaction between individual myofilaments and membrane-bound actin filaments. Here we reconstituted a minimal actin cortex to directly visualize the action of individual myofilaments on membrane-bound actin filaments using TIRF microscopy. We show that synthetic myofilaments fragment and compact membrane-bound actin while processively moving along actin filaments. We propose a mechanism by which tension builds up between the ends of myofilaments, resulting in compressive stress exerted to single actin filaments, causing their buckling and breakage. Modeling of this mechanism revealed that sufficient force (~20 pN) can be generated by single myofilaments to buckle and break actin filaments. This mechanism of filament fragmentation and compaction may contribute to actin turnover and cortex reorganization during cytokinesis.

**\*For correspondence:** svogel@biochem.mpg.de (SKV); schwille@biochem.mpg.de (PS)

**Competing interests:** The authors have declared that no competing interests exist

**Reviewing editor**: Mohan Balasubramanian, Temasek Life Sciences Laboratory, Singapore

## Introduction

The actin cortex consists of a thin actin meshwork bound to the inner cytosolic face of the plasma membrane by various anchor proteins (**Morone et al., 2006**). It plays a pivotal role in providing mechanical stability to the cell membrane, and in controlling cell shape changes during cell locomotion and cell division (**Wessells et al., 1971**; **Bray and White, 1988**; **Diz-Muñoz et al., 2010**; **Sedzinski et al., 2011**). Many of these features rely on proper functioning of the actin motor myosin II (**De Lozanne and Spudich, 1987**; **Cramer and Mitchison, 1995**). Myosin II in the actin cortex functions as assemblies of motor proteins forming anti-parallely arranged bipolar filaments (myofilaments) with motor domains on both filament ends ((**Verkhovsky and Borisy, 1993**; **Verkhovsky et al., 1995**), **Figure 1A**). Besides being the force generator necessary for cell cortex remodeling and actomyosin ring constriction during cytokinesis, there is evidence that myofilaments also contribute to actin filament turnover during cyotkinesis (**Burgess, 2005**; **Guha et al., 2005**; **Murthy and Wadsworth, 2005**). In all these processes, the microscopic mechanism of the interaction between individual myofilaments and membrane-bound actin filaments is not understood. Due to the vast complexity of cellular systems, much effort has been spent to investigate the consequences of actin-myosin interactions from the in vitro perspective (**Backouche et al., 2006**; **Smith et al., 2007**; **Schaller et al., 2010**; **Kohler et al., 2011**; **Soares e Silva et al., 2011**; **Gordon et al., 2012**; **Reymann et al., 2012**). However, these studies focused on myosin-induced actin structure formation on a mesoscopic scale, rather than on the interaction between actin and individual myofilaments. In addition, membrane-bound minimal actin systems have only recently begun to be functionally reconstituted (**Vogel and Schwille, 2012**). To fill the gap in understanding individual myofilament–actin interactions, we directly visualized the action of myofilaments on membrane-bound actin filaments in a minimal in vitro system and complemented the experimental findings with a theoretical model.

**eLife digest** Actin is a multi-functional protein that is found in almost all eukaryotic cells. When polymerized, it forms robust filaments that participate in a variety of cellular processes. For example, actin filaments are involved in the contraction of muscles, and they are also a major component in the various structures that maintain and control the shape of cells as they move and divide. These structures include the cell cortex, a meshwork of actin filaments that is bound to the inner surface of the plasma membrane by anchor proteins. However, both the cell cortex and the plasma membrane must undergo dramatic changes when a cell divides, and the forces that drive these changes are generated by another protein, myosin II.

Myosin II contains three domains: a head domain, also known as the motor domain, that binds to actin; a neck domain; and a tail domain. Like actin, myosin II proteins also form filaments, but these myofilaments have a distinctive structure: the tail domains of two Myosin II proteins join together, with the motor domains being found at both ends of the filament. When activated, the motor domains grab actin filaments and pull against them in a 'powerstroke'. However, the details of the interactions between the myofilament motor domains and the actin filaments in the cell cortex, which are bound to the plasma membrane, are not fully understood.

Studying these processes in living cells is extremely challenging, so Vogel et al. have built an in vitro model of the cell cortex, and then used single-molecule imaging to watch the interactions between the myofilaments and the actin filaments in this model. They show that the myofilaments move along the actin in the cortex, breaking up the filaments and compressing them in the process. They propose that tension builds up between the ends of the myofilaments, leading to compressive stress being exerted on the actin filaments. Computer simulations confirm that the forces generated are high enough to cause the actin filaments to buckle and break. The in vitro model developed by Vogel et al. should allow researchers to clarify the basic biophysical principles that underpin the structure and function of the cell cortex.

## Results

To mimic the cell cortex, we developed a 'minimal actin cortex' (MAC) consisting of actin filaments coupled to a supported lipid bilayer via biotin neutravidin bonds (*Figure 1A*). We used Alexa-488-phalloidin stabilized as well as non-stabilized (data not shown) actin filaments for the MAC composition. By varying the amount of biotinylated lipids present in the lipid bilayer, we have control over the density of the actin layer (*Figure 1B*). In order to understand the origin of contractility observed in cell cortices, we tested the response of the MAC upon addition of myosin motors. Rabbit muscle myosin II was purified and reassembled forming bipolar synthetic myofilaments with a typical length of 500–600 nm (*Figure 1C*). Time lapse imaging of MACs with various actin densities using total internal reflection fluorescence microscopy (TIRFM) showed a dynamic rearrangement of the actin filaments, and subsequently the formation of actomyosin foci in an ATP-dependent manner immediately after addition of myofilaments (*Figure 1D*, *Movie 1*). Actin pattern formation occurred at ATP concentrations between 0.1–1 µM in systems where ATP is enzymatically regenerated (*Table 1*) in 94% of the experiments (n = 45 experiments). At higher and lower ATP concentrations, actin pattern formation in the MAC is absent. Actin structure formation after ATP depletion has been also reported for experiments using actin and myofilaments in solution (*Smith et al., 2007*). At low ATP concentrations, myofilaments were predicted to function as active temporary crosslinkers, which may drive self-assembly of actin filaments into actin clusters.

To understand the details of myofilament–actin interactions at low ATP concentrations during actin pattern formation, we added myofilaments to low density MACs. If crosslinking of actin by myofilaments were the only prerequisite for pattern formation, we would expect no (fast) pattern formation in low density MACs, due to the long distances between the actin filaments in relation to the length of myofilaments. Strikingly, myofilament addition to low density MACs displayed breakage events and compaction of actin filaments, resulting in their shortening over time in all experiments (n = 21, *Figure 2A–C*, *Movie 2*). After 20 min, the majority of the actin filaments have been shortened to, on average, half of their original length, and most of the fragments coalesced into single foci (*Figure 2A,B*, *Movie 2*). Note that actin filaments remained intact when imaged in the absence of myofilaments (data

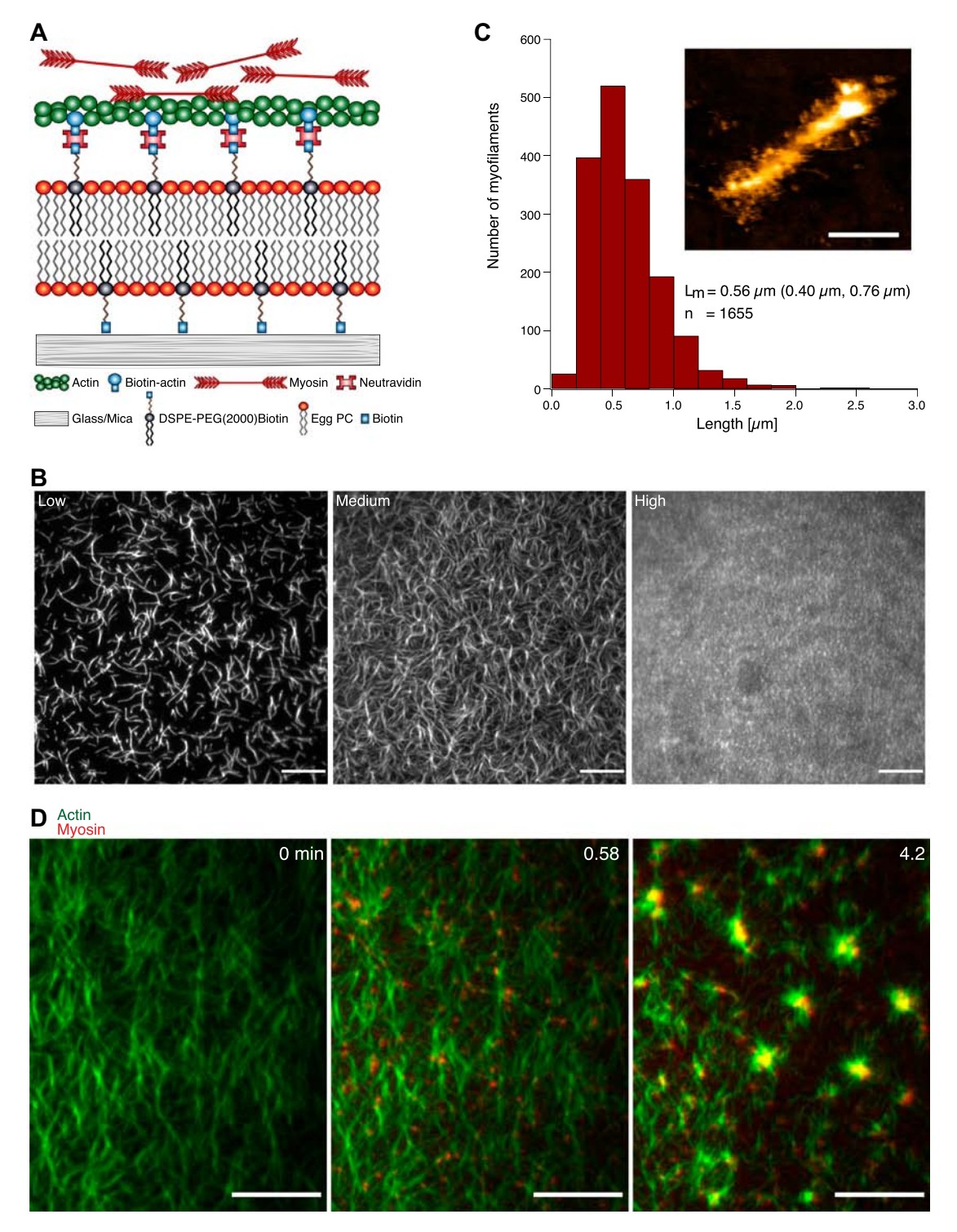

**Figure 1**. MAC composition and actin pattern formation by myofilaments. (**A**) Scheme of the MAC. Biotinylated actin filaments are coupled to a supported lipid bilayer (Egg PC) containing biotinylated lipids (DSPE-PEG(2000)-Biotin) via Neutravidin. (**B**) TIRFM images of MACs containing Alexa-488-phalloidin labeled actin filaments. The increase of actin filament densities (left to right) corresponds to an increase in the amount of DSPE-PEG200-Biotin (low = 0.01 mol%, medium = 0.1 mol%, high = 1 mol%) in the membrane. Scale bars, 10 µm. (**C**) Length distribution of myofilaments. The median length ($L_m$) and the 25th and 75th percentile (brackets) are indicated in µm. Inset shows a topographical AFM image of a myofilament. Height, 12 nm; scale bar 200 nm. (**D**) Dual-color TIRFM time-lapse images of a medium actin density MAC with Alexa-488-phalloidin labeled actin filaments (green) and myofilaments (0.3 µM unlabeled myosin II doped with Alexa 647 myosin II (red)) before (left image) and during actin pattern formation. Scale bars, 10 µm.

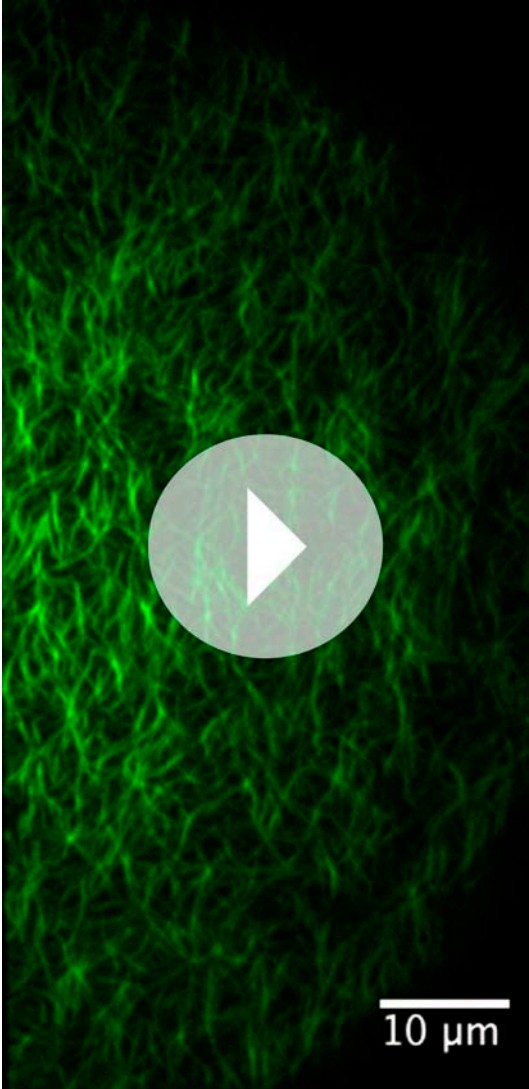

**Movie 1**. Actin pattern formation by myofilaments.

not shown). Upon myofilament addition, actin filaments frequently showed deformations prior to actin filament breakage (*Figure 2C*, yellow arrowheads, *Movie 3*), indicating that force is exerted by the myofilaments and stress along the actin filament may build up until the actin filament breaks. Similarly, recent evidence implied that exposure of actin/fascin bundles to myofilaments can induce their disassembling and severing by an unknown process (*Backouche et al., 2006*; *Haviv et al., 2008*; *Thoresen et al., 2011*).

Furthermore, an increase in fluorescence intensity along the remaining actin filament is often observed after a fragmentation event (*Figure 2C*, white arrowheads, *Figure 2D,E*). The fluorescence intensity proximal to the breakage site is approximately twice as high compared to the rest of the actin filament, suggesting that the fragment is dragged along the remaining actin filament by the myofilaments leading to its compaction (*Figure 2C–E*, *Movie 3*). We propose that fragmentation and compaction contribute to the observed coalescence of actin fragments into single foci during the dynamic rearrangement of the actin filaments (*Figures 1D and 2A*, *Movies 1 and 2*).

To determine how myofilaments execute fragmentation and compaction and to test whether these processes demand (concerted) actions of a multitude of myofilaments, we reduced the myofilament concentration to the single molecule level. Alexa-647 labeled myofilaments were added to MACs with low actin density and imaged by two-color TIRFM. Upon binding of single myofilaments to individual actin filaments, we observed directed movement of myofilaments along actin filaments and actin fragmentation at low ATP concentrations (*Figure 3A*, *Movie 4*, *Table 1*, *Figure 5*). 68% of the individually observed myofilaments displayed directed movement and 50% exhibited fragmentation and compaction of an actin filament (total number of myofilaments = 152; 7 experiments). In cases where both fragmentation and compaction occurred, 95% of the observed myofilaments showed directed movement along the actin filaments, while 5% remained stationary at their original binding site (total number of fragmenting myofilaments = 75; seven experiments).

In contrast, at high ATP concentrations, processive movement of myofilaments was barely visible, and fragmentation of actin filaments was absent (supplementary text in 'Material and methods', *Figure 5*, *Movie 5*).

Shortly after binding of myofilaments, actin filaments deformed and eventually broke, indicating that the force generated by a single myofilament is sufficient to break an actin filament (*Figure 3A*, yellow arrowheads, *Movie 4*). Subsequently, the fluorescence signal of actin filaments proximal to the breakage site increased (*Figure 3A*, white arrowheads, *Figure 3C*). The detected increase in the fluorescence intensity of actin filaments implies that actin fragments are further dragged along the remaining actin filaments by the myofilaments during their movement (*Figure 3A,D*, *Movie 4*). We analyzed our TIRFM data in greater detail by tracking single myofilaments during their directed movement along an actin filament (*Figure 3B*, (*Rogers et al., 2007*)) and determining the velocity from the trajectories (*Figure 3C*, red curve). Simultaneously, the fluorescence intensity in the green (actin) channel of the area occupied

**Table 1.** ATP dependency of contraction and fragmentation. ATP was kept at a constant level during the experiment by enzymatic regeneration (see 'Material and methods'). Buffer containing ATP concentrations listed in the table and 0.3 µM myofilaments were added to medium and or low-density MACs. Contraction here is defined as visible dynamic rearrangements of actin filaments after myofilament addition. Fragmentation implies visible actin filament breakage events after myofilament addition

| Regenerated ATP concentration (µM) | Contraction | Fragmentation |
|---|---|---|
| 100 | No | No |
| 50 | No | No |
| 25 | No | No |
| 12.5 | No | No |
| 10 | No | No |
| 1 | Yes | Yes |
| 0.3 | Yes | Yes |
| 0.1 | Yes | No |
| 0 | No | No |

by the myofilament during its movement was measured, indicating breakage events as the dragged fragment leads to an increase of fluorescence proximal to the breakage site (*Figure 3C*, blue and black curve). The velocity fluctuates during the movement of the myofilament along the actin filament (*Figure 3C*, red curve), whereby acceleration events (*Figure 3C*, red arrowheads) are closely followed by an increase in the actin fluorescence, indicating that a breakage event occurred (*Figure 3C*, black arrows, blue and black curve). We hypothesize that phases of increasing tension between the ends of the myofilament lead to deformation of the actin filament and coincide with a decrease in velocity, while the phases of tension release immediately after actin filament breakage may result in acceleration of the myofilament (*Figure 3C,D*).

In order to explain the buckling and eventual breakage of the actin filament, we propose the following model: The myosin filament aligns parallel to the actin filament and interacts with it via the myosin heads ((*Sellers and Kachar, 1990*), *Figure 3D*). One end of the myofilament, which we will refer to as the 'leading end', is oriented towards the actin plus end (barbed end), as the filaments in a muscle, and can therefore walk along the actin filament while hydrolyzing ATP (*Figure 3D*). The other end of the myofilament, which we will refer to as the 'trailing end', separated from the leading end by the approximately 160 nm long bare zone (*Al-Khayat et al., 2010*) without myosin heads, is oriented in the opposite direction. While the trailing end still interacts with the actin filament, its opposite orientation results in a much slower directed motion towards the actin plus end (*Spudich et al., 1985*; *Sellers and Kachar, 1990*). Every myosin head independently follows the biochemical cycle consisting of ATP hydrolysis, binding to actin, powerstroke accompanied by phosphate dissociation, ADP dissociation, ATP binding, and finally detachment from actin ((*Howard, 2001*), *Figure 8*). We assume that the heads on the trailing end still interact with the actin filament and go through the same cycle, but either do not perform steps that would lead to a processive motion (probability of making a step $p_{st}$=0), or make steps with a small probability ($p_{st}$=0.1). When a myosin head makes a step (step size d = 5 nm (*Howard, 2001*)) it tries to move the whole myosin filament towards the actin plus end, but other attached myosin heads are holding it back, thus generating friction (*Figure 3D*). The myofilament trailing end functions mainly as a source of friction (an effective 'brake'), but is also moved towards the plus end by the pulling force exerted by the leading end (*Figure 3D*). Since mainly the leading end is actively moving and the trailing end is either passively pulled ($p_{st}$=0) or contributes only weakly to its own motion ($p_{st}$=0.1), tension builds up within the myofilament, and is transferred onto the actin filament as a compressive force. If sufficiently high, this compressive force can cause buckling, and finally breakage, of the actin filament (*Figure 3D*). After the breakage, the leading end can move unhindered further towards the plus end, while dragging along the broken-off part of the actin filament attached to the trailing end (*Figure 3D*). The trailing end can also attach again to the actin and further breakage events on the same filament can follow.

In order to estimate the force needed to break an actin filament, we model the filaments as flexible rods with bending rigidity EI = 60 nN µm$^2$, determined from the persistence length of actin: $l_p$ = EI/(kT) = 15 µm (*Yanagida et al., 1984*). The force needed to buckle and break a filament is F = $\pi^2$ EI/$l^2$. With the length l of the myofilament bare zone of 160 nm, this gives a force of 23 pN. Can the tension within the myofilament reach up to this force? We performed simulations of this model, describing the attached myosin head as a spring with a spring constant equal to the myosin head stiffness κ = 1 pN nm$^{-1}$ (*Kaya and Higuchi, 2010*), and equilibrate all forces after every step and every detachment of a myosin head. From the AFM image of the myofilament (*Figure 1C*) and the known myofilament

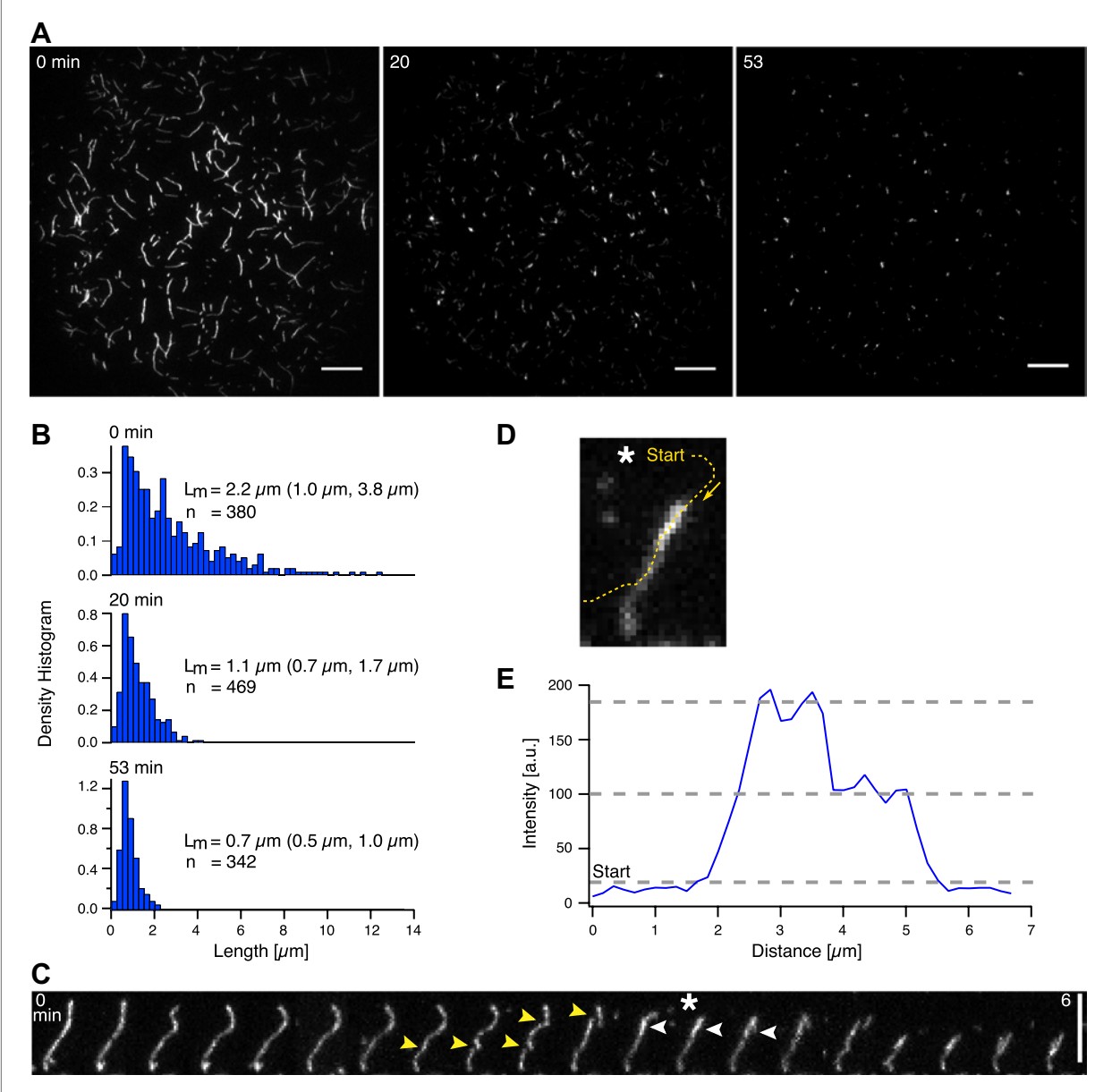

**Figure 2**. Actin filament shortening and compaction by myofilaments. (**A**) TIRFM time-lapse images of a low actin density MAC with Alexa-488-phalloidin labeled actin filaments before (left image) and after addition of (non-labeled) myofilaments (0.3 μM). Scale bars, 10 μm. (**B**) Actin filament length distribution at 0, 20 and 53 min after myofilament addition. The median length ($L_m$) and the 25th and 75th percentile (brackets) are indicated in μm. (**C**) TIRFM time-lapse sequence of an Alexa-488-phalloidin labeled actin filament in the presence of myofilaments (0.3 μM). Yellow arrowheads point at deformation and breakage events. White arrowheads indicate an increase in fluorescence intensity. Scale bar, 5 μm. (**D**) and (**E**) image and the corresponding intensity profile (blue curve) of the actin filament. The intensity was measured along the yellow dashed line shown in (**D**). The line started and ended outside the actin filament to indicate the background level. Asteriks in (**C**) and (**D**) mark the image taken for the intensity profile measurement.

structure (***Woodhead et al., 2005***), we estimated that there are $n_m = 30$ interacting myosin heads per filament. The result is a net movement of the whole myofilament towards the actin plus end, with fluctuating tension force, velocity and number of attached myosin heads, the mean values of which depend on the ATP concentration (***Figures 4A,B and 6***). We note that the tension force increases with decreasing ATP concentration, reaching forces necessary for breakage only at low (lesser than ~3 μM) ATP concentrations, in agreement with the experiments (***Figure 4A***, see also ***Table 1***). When the actin

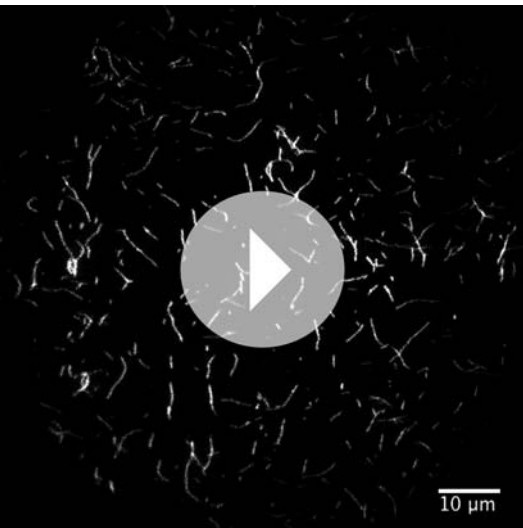

**Movie 2**. Shortening of individual actin filaments by myofilaments.

filament was allowed to bend in simulations at the point where the buckling force of 23 pN was reached (**Figure 4B**), thus releasing the excess tension, the filament curvature steadily increased at low ATP concentrations, reaching the critical curvature of breaking of 5.6 μm$^{-1}$ ((**Arai et al., 1999**), **Figure 7A**). At intermediate ATP concentrations, the actin curvature fluctuated, at times exceeding the curvature threshold (**Figures 4C and 7B**). At higher ATP concentrations, the threshold force was reached for too short periods for the filament to be bent to the breakage point (**Figures 4D and 7C**). The results do not depend significantly on whether the myosin heads on the trailing end perform steps ($p_{st}$=0.1) or not ($p_{st}$=0). The simulations thus support the idea that in the ATP concentration range used in the experiments, the differences in the interactions of the trailing and leading ends of the myofilament with actin can generate compressive forces on the actin filament, and that these forces are sufficiently high to bend and break the actin filament.

## Discussion

In our in vitro study we provide a potential mechanism how actin turnover in cells may be mediated by myofilament driven actin fragmentation. To bend and break the actin filament, the myofilament has to be attached to the actin for sufficient time, and sufficient force has to be generated, requiring a certain mean number of heads being attached to actin at any given time. Given the small number of interacting myosin heads this translates to a requirement that a large fraction of heads is in the bound state. In our assay this was achieved with low ATP concentrations. It is tempting to speculate that actin breakage by myofilaments in vivo is governed by the ATP level in the cell. Recent evidence exists that ATP levels indeed vary significantly inside living cells (**Imamura et al., 2009**). However, physiological ATP concentrations, derived from methods that provide averaged ATP levels with no high spatial and temporal resolution from cell extracts, are usually found in the millimolar range (**Beis and Newsholme, 1975**). Here it is important to mention that skeletal muscle myosin II (used in this study) has a lower duty ratio and is therefore less processive than non-muscle myosin in cells (**Harris and Warshaw, 1993**; **Wang et al., 2003**). By lowering the ATP concentration in our assay, we increased the duty ratio of our myofilaments and thereby made them more processive (supplementary text in 'Material and methods', **Figure 5**, **Movie 5**), similar to non-muscle myosin, which is in line with previous studies (**Humphrey et al., 2002**; **Smith et al., 2007**; **Soares e Silva et al., 2011**). Moreover, processivity of myofilaments in animal cells may be also controlled by phosphorylation of myosin light chains through other proteins that

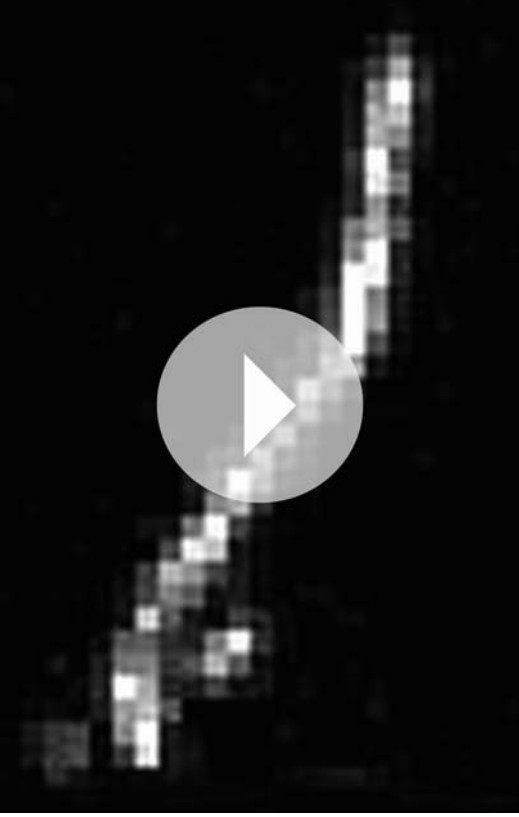

**Movie 3**. Fragmentation of a single actin filament.

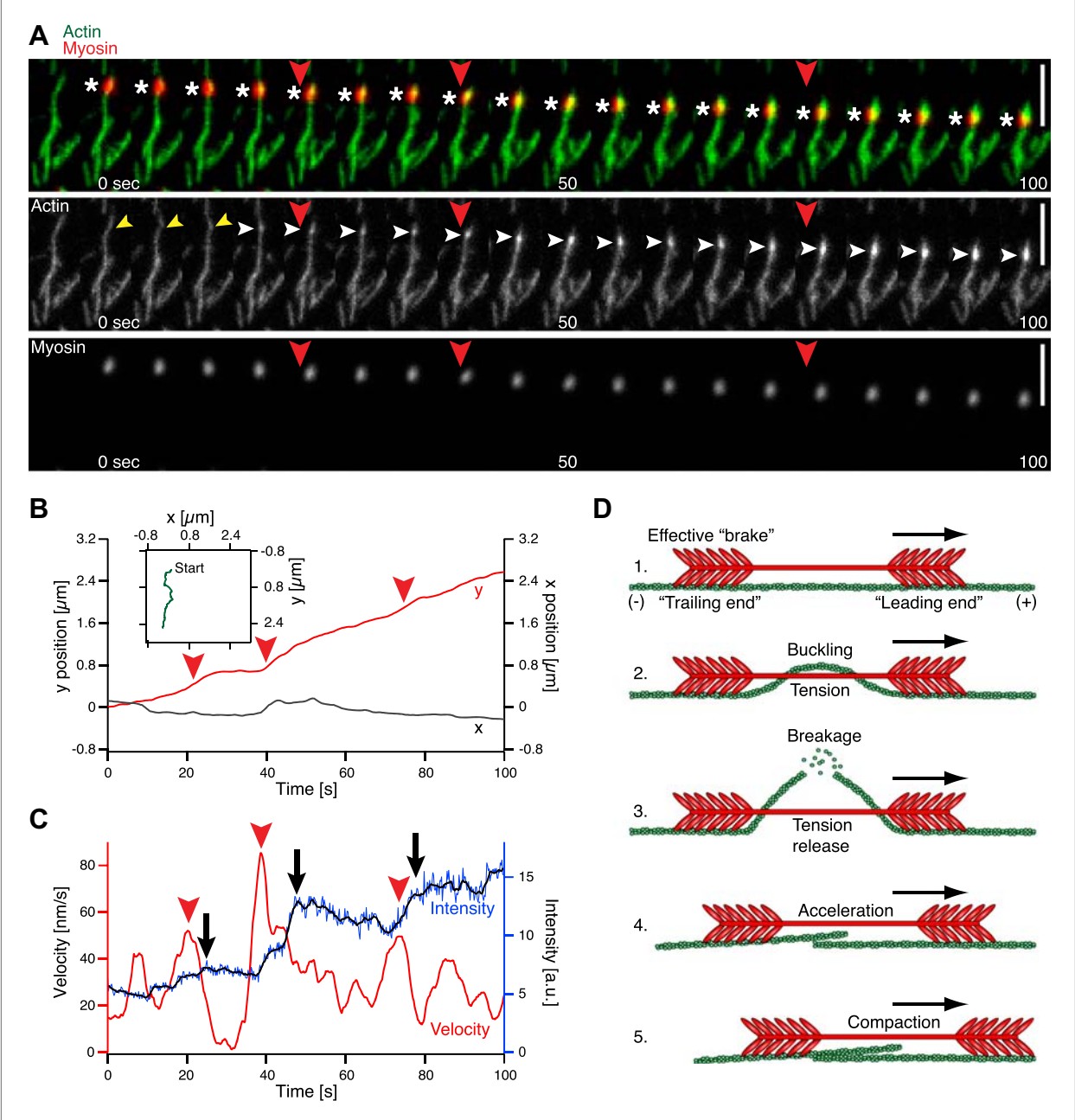

**Figure 3**. Single molecule analysis of the myofilament movement and actin fragmentation. (**A**) Dual-color TIRFM time-lapse sequence of a Alexa-647 labeled myofilament (red) moving along an Alexa-488-phalloidin labeled actin filament (green). White asterisks mark the position of the myofilament. Yellow arrowheads point to actin filament deformations. White arrowheads indicate an increase in fluorescence intensity. Scale bars, 5 μm. (**B**) x (grey curve) and y (red curve) positions of the myofilament movement shown in (**A**) as a function of time. Inset depicts the trajectory (green curve). (**C**) Myofilament velocity (red curve) calculated from the xy positions in (**B**) and actin filament intensity (blue [raw data] and black [smoothed] curves) over time. Red arrowheads denote acceleration events. Black arrows point to fluorescence intensity increases. Red arrowheads in (**A**)–(**C**) mark corresponding time points in (**A**). (**D**) Proposed model for myofilament driven actin fragmentation and compaction (details in text).

modify the kinetic rates of the biochemical cycle and thereby increase the duty ratio of the myosins shifting the observed behavior to a region of high physiological ATP concentrations (*Tan et al., 1992*; *DeBiasio et al., 1996*; *Matsumura et al., 2001*).

Our simulations show that the model of interaction between the actin and myosin filaments described in the text is plausible given the quantitative parameter values known from literature (rate constants).

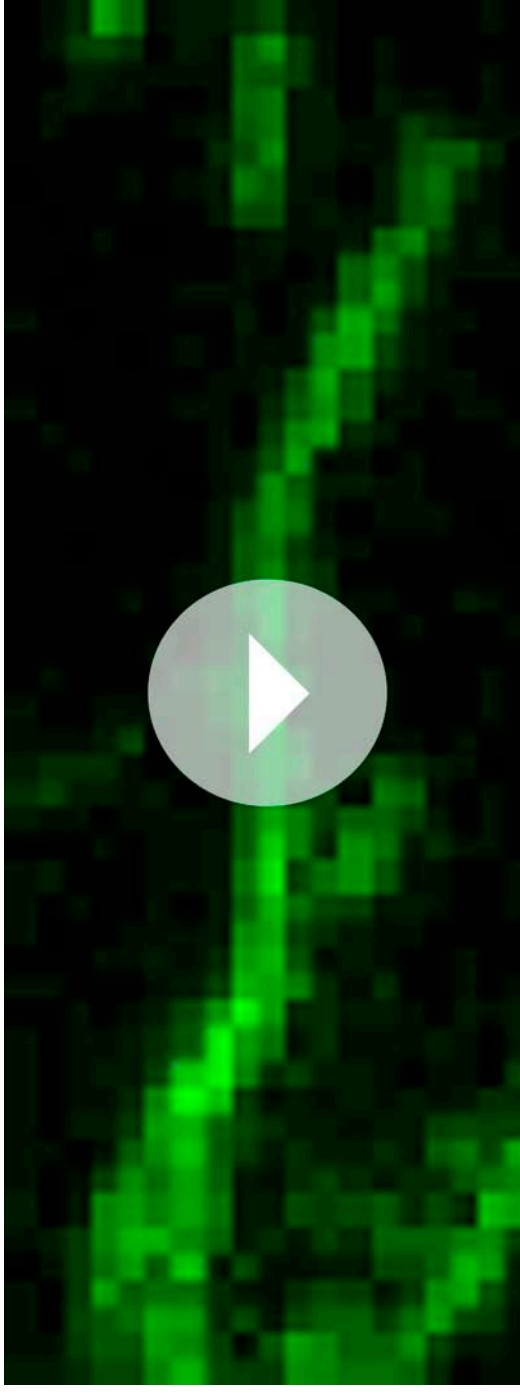

**Movie 4**. Movement and actin fragmentation by a single myofilament.

Most importantly, it shows that sufficient force needed to bend and break the actin filament can be generated by a single myofilament, assuming only the difference in the interaction between the leading and trailing myofilament ends (making a step vs not making a step). This result implies that neither actin nor myosin attachment to a support or any other rigid structure is necessary for the observed process. The forces act within a single myofilament. Binding of the actin filaments to the membrane provides confinement of the filaments to a plane, without rigidly tethering the filaments to a support or a scaffold. The motion of actin is only restricted by the effectively higher viscosity of the membrane compared to the buffer solution. The simulation further yields quantitative details of the model, for example, the mean fractions of myosin heads in the six possible states (*Figure 9*), and describes the fluctuations of the actin filament curvature (*Figure 4C,D*). Other parameters that can be derived from the simulations and could be compared with new experiments include the processivity of myosin, as a function of the ATP concentration, and the dependence of the overall behavior on the length of myofilaments.

In conclusion, our findings show the distinct functions that myosin motors can execute. In our minimal system, myofilaments fragment and compact membrane-bound actin. We directly show that single myofilaments can interact with actin in such a way that sufficient force can be generated to break the filament, without either the myofilament or the actin being firmly attached to a solid support or scaffold. We suggest that fragmentation and compaction by myofilaments contributes to the observed large-scale pattern formation of actomyosin networks also in other in vitro systems (*Backouche et al., 2006*; *Smith et al., 2007*; *Soares e Silva et al., 2011*; *Gordon et al., 2012*). In vivo breakage of actin bundles has been shown to occur in neuronal growth cones in a myosin II dependent manner, and is thought to be important for recycling actin (*Medeiros et al., 2006*). We propose that the observed fragmentation and compaction of membrane-bound actin filaments by myosins in our in vitro system may serve as a possible general mechanism for actin turnover and actin cell cortex remodeling.

## Material and methods

### Details of the experiments

Traditionally, single muscle myosin II motors are described as non-processive motors (*Howard, 2001*). By assembling myosin motors into filaments, the myofilaments become processive due to the coupling of a higher number of myosin heads that are in contact with the actin filament. In our minimal system directed movement accompanied by actin filament fragmentation only occurred at low ATP

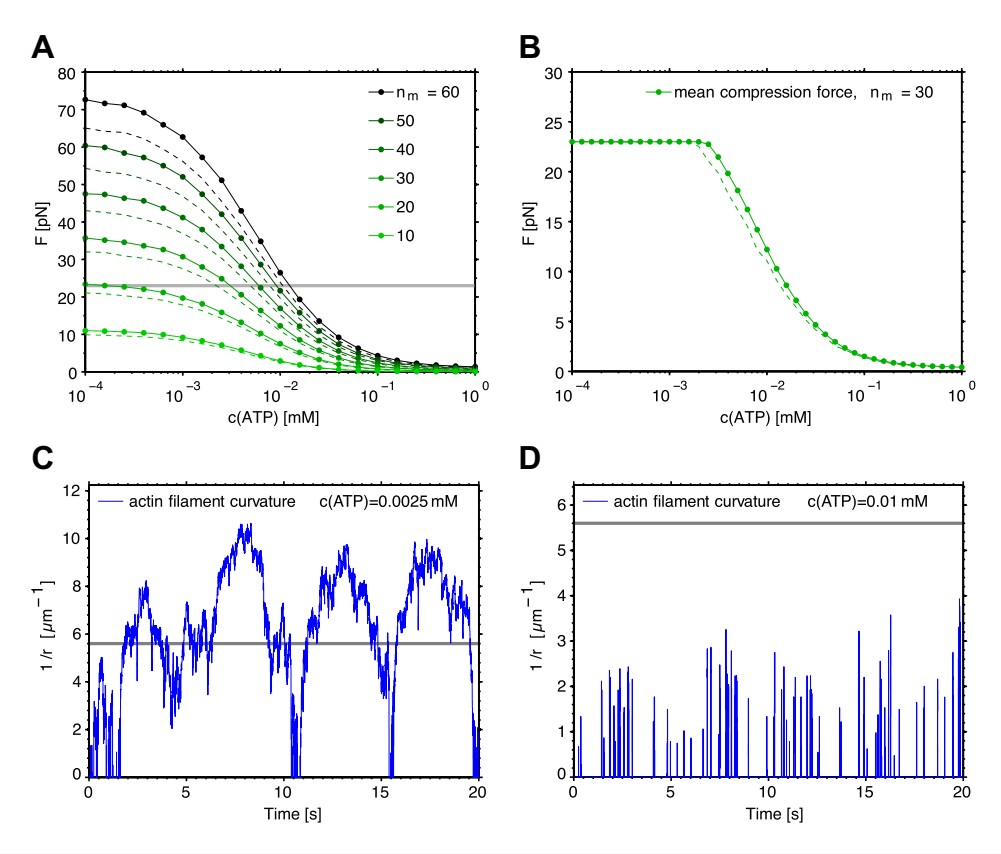

**Figure 4**. Simulation of the interaction between myofilaments and an actin filament. (**A**) Mean tension force F within the myofilament when bending of actin is not allowed; dependence on ATP concentration for several different numbers of interacting myosin heads $n_m$. The forces when the myosin heads of the trailing end are not performing steps ($p_{st}$=0, points connected by a solid line) are slightly higher than the forces when the steps occur with the probability $p_{st}$=0.1 (dashed lines). (**B**) Mean tension force when the actin filament is allowed to bend at the threshold force of 23 pN (points and solid line: $p_{st}$=0, dashed line: $p_{st}$=0.1). (**C**) Actin filament curvature fluctuations during 20 s of the simulation at 0.0025 mM ATP concentration, showing that the critical curvature of 5.6 µm$^{-1}$ needed for actin filament breakage is often reached, while at higher ATP concentration (0.01 mM), the critical curvature is never reached (**D**) ($p_{st}$=0 in (**C**) and (**D**)).

concentrations between 0.1–1 µM (*Figure 3*, *Table 1*, *Movie 3*). The lower ATP concentrations are expected to increase the duration of the actin-bound post-working stroke state of the myosin head, thus increasing the duty ratio and processivity (*Howard, 2001*). Extremely low ATP concentrations are on the other hand not sufficient for myosin motor activity. To check the effect of ATP concentration on processivity we added Alexa-647 labeled myofilaments to medium density MACs and tracked (*Rogers et al., 2007*) the movement of individual myofilaments at low (1 µM) and high (4 mM) ATP concentrations (*Figure 5*, *Movie 5*). (Note that fragmentation and compaction of actin filaments only occurred at low ATP concentrations). Comparison of the trajectories illustrates that low ATP trajectories are on average longer than high ATP trajectories indicating a higher processivity at low ATP conditions (*Figure 5*, *Movie 5*). Moreover the number of tracked myofilaments at low ATP concentration with a dwell time greater than 900 ms was more than six times higher than at high ATP concentration suggesting a higher duty ratio at low ATP levels (n = 681 at low ATP, n = 102 at high ATP; *Figure 5*, *Movie 5*). High ATP concentrations therefore lead to a faster detachment of myofilaments from actin filaments than at low ATP concentrations (*Movie 5*). We propose that the increase in processivity due to ATP deprivation is necessary for the processive movement and a prerequisite for fragmentation and compaction of actin filaments.

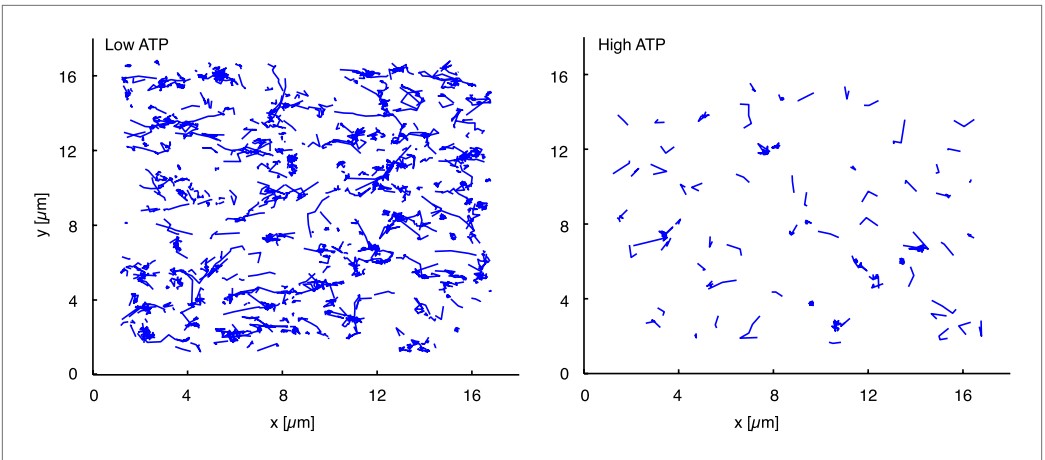

**Figure 5**. Trajectories of individual myofilaments moving along actin filaments. Left panel displays the trajectories of myofilaments at low (1 µM) ATP concentration (see also the corresponding *Movie 5*). Right panel shows trajectories of myofilaments at high (4 mM) ATP concentration (see also the corresponding *Movie 5*). Myofilaments were tracked for one minute and those who stayed less than 900 ms attached to the actin filament were filtered out (*Rogers et al., 2007*).

## Details of the model

### Biochemical cycle

In the model of the myofilament interaction with actin, we assume that each active myosin head goes through the following biochemical cycle consisting of six head states (*Howard, 2001*). The six states are schematized in *Figure 8*. Unbound myosin head with ATP (state 1) hydrolyses ATP with the rate $k_1 = 100$ s$^{-1}$ (state 2), binds to the actin filament with the rate $k_2 = 30$ s$^{-1}$ (state 3), rapidly releases phosphate while performing a powerstroke with rate $k_3 = 10^4$ s$^{-1}$ (state 4), releases ADP with the rate $k_4 = 1000$ s$^{-1}$ (state 5), binds ATP with the rate $k_5 = k_t$ [ATP], where $k_t = 4$ µM$^{-1}$ s$^{-1}$, (state 6), and dissociates from the actin filament with the rate $k_6 = 2000$ s$^{-1}$ (state 1). These rates determine the average time the myosin heads spend in every state (*Figure 9*). In this simple model the rates are assumed to be independent of the strain of the myosin head. The myosin step size is d = 5 nm (*Howard, 2001*). The myosin heads of the leading myofilament end always perform a step as a result of the powerstroke between the states 3 and 4. The heads of the trailing end either do not make a step (the probability of making a step $p_{st}$=0), or, in separate simulations, make a step with the probability $p_{st}$=0.1. This value is based on the observations that the trailing myofilament end moves along actin with approximately ten-times slower speed than the leading end (*Sellers and Kachar, 1990*).

### Mechanical equillibrium

Every myosin head attached to the actin filament is described as a spring with a spring constant equal to the myosin head stiffness κ = 1 pN nm$^{-1}$ (*Kaya and Higuchi, 2010*). After every step and after every detachment of a myosin head the forces are equilibrated by moving the myofilament along the actin filament. The result is a net movement of the whole myofilament towards the actin plus end, with flucuating tension force, velocity and number of attached myosin heads, the mean values of which depend on the ATP concentration.

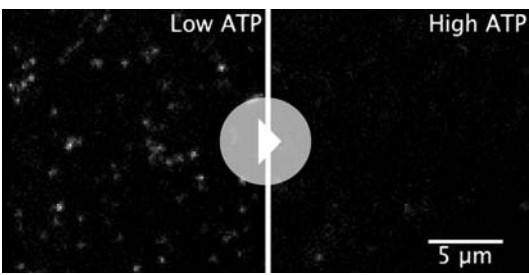

**Movie 5**. Behavior of single myofilaments at low and high ATP concentrations.

### Number of myosin heads on the myofilament

The median length of the myofilaments determined with AFM was 560 nm (*Figure 1C*). Assuming the length of the bare zone without myosin heads in the central part of the myofilament to be 160 nm (*Al-Khayat et al., 2010*), both ends of the

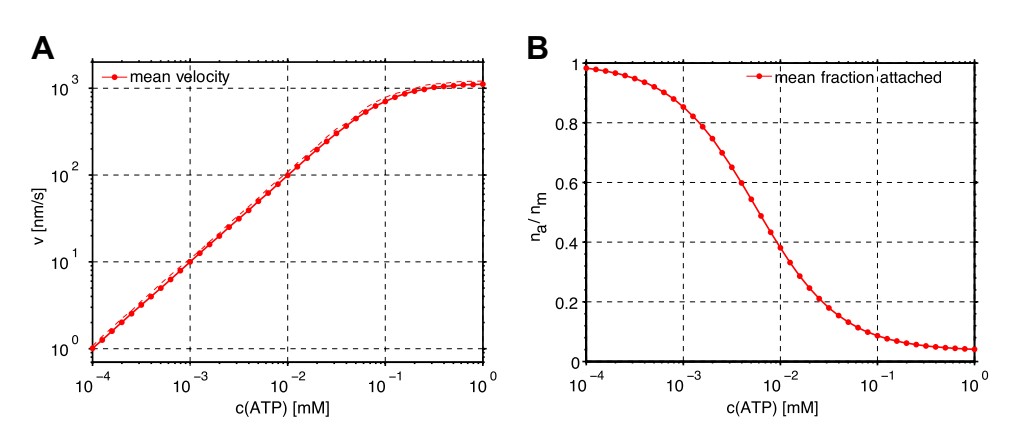

**Figure 6**. Dependence of the mean velocity and mean number of attached myosin heads on the ATP concentration obtained from the simulations of myofilament–actin filament interaction. (**A**) Mean velocity (points and solid line: $p_{st}=0$, dashed line: $p_{st}=0.1$). (**B**) Mean fraction of attached myosin heads.

myofilament are 200 nm long. With four myosin head pairs around the myofilament circumference per every 14.5 nm (**Woodhead et al., 2005**), there are on average 110 head pairs on each myofilament. Assuming further that only those heads oriented towards one side, that is, one quarter, can interact with the actin filament, and that only one head of the head pair is favorably oriented to interact, we estimate the average number of interacting heads as 30 per myofilament.

## Buckling force

In order to estimate the force needed to bend the actin filament, we model it as a flexible rod with bending rigidity EI = 60 nN µm$^2$, determined from the persistence length of actin: lp = EI/(kT) = 15 µm (**Yanagida et al., 1984**). The force needed to buckle and break the filament is F = π$^2$ EI/l$^2$. With the length l of the myofilament bare zone of 160 nm, this gives a force of 23 pN. Actin filaments break when the radius of curvature of a bent filament decreases below 0.18 µm (**Arai et al., 1999**) corresponding to the curvature 1/r = 5.6 µm$^{-1}$.

## Results of the simulations

In order to find out if the tension within the myofilament transferred onto the actin filament as a compressive force can become sufficiently high to bend and break the actin filament, we performed two types of simulations.

In the first case, bending of the filament was not allowed and no limit was imposed on the compression force within the actin filament. This allowed us to determine the forces that can be reached by this model. In the second case, the actin filament was allowed to bend when the force of 23 pN was exceeded. Actin bending was modeled by decreasing the distance between the leading and trailing ends of the myofilament, thus relaxing the stress and reducing the force down to 23 pN.

In case when bending was not allowed, the average force increased with decreasing ATP concentration, and also with the increasing number of myosin heads (**Figure 4A**). For 30 myosin heads, the force of 23 pN, neccessary to bend the filament, could be reached at ATP concentrations of approximately 3 µM or lower (**Figure 4A**).

In the ATP concentration range used in the experiments here (<100 µM, see also **Table 1**) the average velocity of myofilament increased approximately linearly with ATP concentration, independently of the number of myosin heads, or of the fact whether actin bending was allowed or not (**Figure 6A**). The average fraction of myosin heads attached to actin increased with decreasing ATP concentration (**Figure 6B**), and since the myosin heads act independently of each other and of the forces involved, was also independent of number of myosin heads and actin bending.

When the actin filament was allowed to bend, the maximum force was not permitted to exceed 23 pN (**Figure 4B**). At low ATP concentration, the generated force was sufficiently high to continuously bend the actin filament, steadily increasing the curvature above the breakage threshold (**Figure 7A**). At

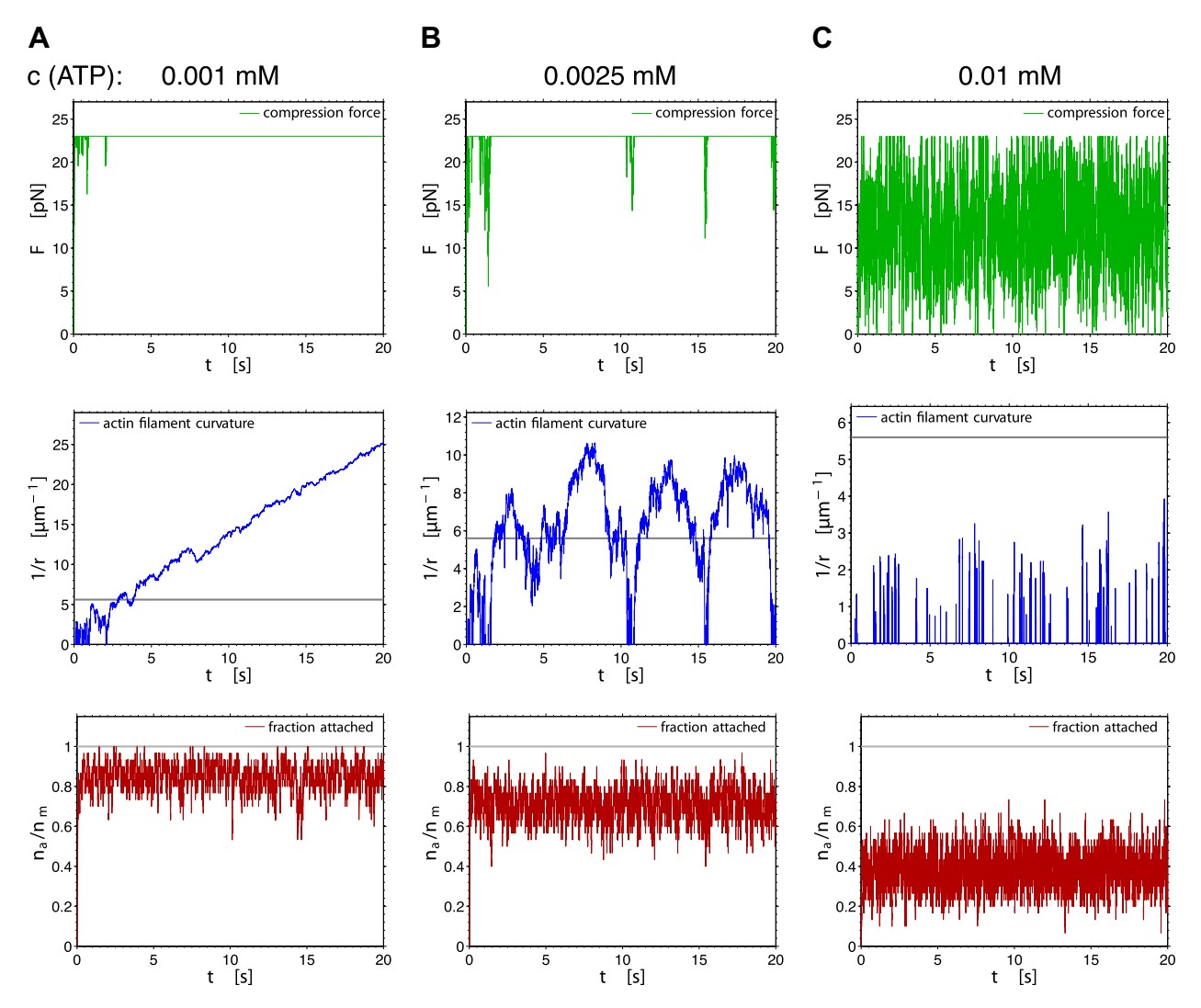

**Figure 7**. Fluctuations of the compression force (upper row), the actin curvature (middle row) and the fraction of attached myosin heads (lower row) during 20 s of the simulation, for three different ATP concentrations ($p_{st}$=0).

intermediate ATP concentrations, the force fluctuating near the threshold caused fluctuations in curvature, at times exceeding the curvature threshold and then relaxing again to a straight filament configuration (*Figures 4C and 7B*). At even higher ATP concentrations, the threshold force was reached for too short periods for sufficiently high curvature to develop (*Figures 4D and 7C*).

The results for the scenario where the myosin heads of the trailing end can make a step towards the plus actin end with the probability $p_{st}$=0.1 are very similar to those with $p_{st}$=0. The differences are that the generated forces are slightly lower (*Figure 4A,B*) and the mean velocity slightly higher (*Figure 6A*).

## Actin preparation and labeling

Rabbit skeletal muscle actin monomers (Molecular Probes) and biotinylated rabbit actin monomers (tebu-bio [Cytoskeleton Inc.]) were mixed in a 5:1 (actin:biotin-actin) ratio. Polymerization of the mixture (39.6 µM) was induced in F-Buffer containing 50 mM KCl, 2 mM MgCl$_2$, 1 mM DTT, 1 mM ATP, 10 mM Tris–HCl buffer (pH 7.5). The biotinylated actin filaments were labeled with Alexa-Fluor 488 Phalloidin (Molecular Probes) according to the manufacturer protocol. Finally, 2 µM (refers to monomers) of Alexa-488-Phalloidin labeled biotinylated actin filaments were obtained.

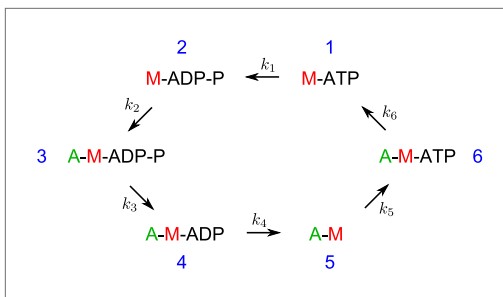

**Figure 8**. The biochemical cycle of the myosin heads with rates $k_1$–$k_6$ assumed in the model and the simulations. The rate $k_5$ is ATP-dependent.

## Myosin preparation and labeling

Myosin was purified from rabbit skeletal muscle tissue as previously described (*Smith et al., 2007*). Myosin activity was tested by a classical motility assay where myosins bound to a nitrocellulose coated glass surface of a perfusion chamber (tebu-bio [Cytoskeleton Inc.]) propel actin filaments. Movement of actin filaments indicated integrity of the myosin motors (data not shown). Myofilament assembly was induced in reaction buffer containing 50 mM KCl, 2 mM MgCl$_2$, 1 mM DTT and 10 mM Tris–HCl buffer (pH 7.5), and depending on the experiments various amounts of regenerated ATP and an oxygen scavenger system. Equilibration of the mixture for approximately 30 min gave us a median length of 560 nm in our system (*Figure 1C*).

Myosins were labeled with thiol-reactive dyes AlexaFluor 488 C$_5$-maleimide as well as AlexaFluor 647 C$_2$-maleimide (both Molecular Probes). The labeling reactions were performed in a slight variation according to the manufacturer protocol. In brief, the thiol-reactive dyes were dissolved in DMSO to 10 mM concentration and stored at −80°C. The myosin stock (14.99 µM in 50% glycerol) was diluted to 2 µM in reaction buffer containing 50 mM KCl, 10 mM Tris–HCl buffer (pH 7.5) and 2 mM MgCl$_2$. The solution was deoxygenated for 15 min under vacuum and put into N$_2$ environment. 15 times molar excess (30 µM) of TCEP (*tris*(2-carboxyethyl)phosphine, Molecular Probes) was added to the solution and incubated for 1 hr at room temperature. 25-fold molar excess to reach 50 µM of the maleimide dyes were added dropwise to the solution while it was stirred and incubated overnight at 4°C. Labeled myofilaments were separated from the remaining dyes by gel filtration to obtain 1 µM (refers to single myosins) labeled myofilaments. Activity of labeled myosins was confirmed by motility and actin pattern formation assays. Aliquots were frozen and stored at −80 °C.

## MAC (Minimal actin cortex) preparation

For MAC preparation a chamber consisting of a cut 1.5 ml Eppendorf tube glued to an air plasma cleaned glass cover slip (22 × 22 mm, #1.5, Menzel Gläser, Thermo Fisher, Braunschweig, Germany) was built. For planar, glass supported lipid bilayer formation molar ratios of the lipids (purchased from Avanti Polar Lipids, Alabaster, AL) Egg PC (99.99, 99.9 and 99 mol%) and DSPE-PEG(2000)-Biotin (0.01, 0.1 and 1 mol%) were dissolved in chloroform (total 10 mg/ml of lipids), dried under nitrogen flux for 30 min and subsequently put into vacuum for 30 min. Lipids were then rehydrated in reaction buffer containing 50 mM KCl, 2 mM MgCl$_2$, 1 mM DTT and 10 mM Tris–HCl buffer (pH 7.5) and resuspended by vigorous vortexing. To obtain SUVs (small unilamellar vesicles) the suspension was exposed to sonication in a water bath at room temperature. 10 µl of the suspension were mixed with 90 µl reaction buffer and placed onto the glass cover slip of the chamber. CaCl$_2$ to a final concentration of 0.1 mM was added to induce fusion of the SUVs and the formation of a lipid bilayer on the glass surface. The sample was washed several times with a total volume of approximately 2 ml reaction buffer in order to remove unfused vesicles. After washing, 2 µg of Neutravidin (Molecular Probes) dissolved in 200 µl reaction buffer was added to the sample and incubated at room temperature for 10 min. The

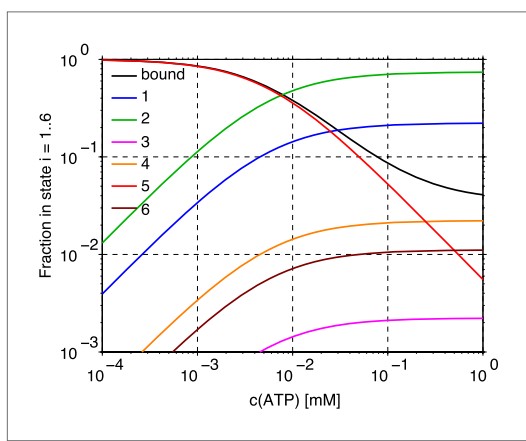

**Figure 9**. The fractions of myosin heads in states 1–6, and in the actin-bound state (sum of states 3–6) in dependence on the ATP concentration. The values are calculated from the model of the myosin head cycle in *Figure 8*.

sample was washed several times with >2 ml reaction buffer to remove unbound Neutravidin. Then 10–50 µl of 2 µM (refers to monomers) Alexa-488-phalloidin labeled biotinylated actin filaments were added to the lipid bilayer and incubated for 1 hr. The sample was carefully washed with approximately 1–2 ml reaction buffer to remove unbound actin filaments.

## Actin fragmentation and pattern formation assay

Alexa-647 (and Alexa-488) labeled myofilaments and/or non-labeled myofilaments of various concentrations (as indicated) dissolved in 200 µl reaction buffer were added to the MAC and imaged by TIRF microscopy. The reaction buffer contained 0.1–1 µM ATP (see also *Table 1*), an ATP regenerating system consisting of 20 mM Creatine phosphate (Sigma) and 0.1 mg ml$^{-1}$ Creatine phospho kinase (Sigma) to keep the ATP concentration constant and an oxygen scavenger system (glucose oxidase (165 U ml$^{-1}$) catalase (2,170 U ml$^{-1}$), β-D-glucose (0.4% wt/vol) and Trolox (2 mM), all from Sigma) to reduce photobleaching of the Alexa dyes. Actin rearrangements and fragmentation occurred immediately after addition of the myofilaments.

## TIRF microscopy

Two color TIRF microscopy was carried out on a custom-made setup built around an Axiovert 200 microscope (Zeiss), for details see (*Loose et al., 2011*). A α Plan-Apochromat 100×/NA 1.46 oil immersion objective and 488 nm and 647 nm laser lines were used for excitation of the labeled probes. The exposure times were either 50 ms or 100 ms, and the time intervals between each recorded frame ranged from 200–400 ms for different experiments.

## AFM imaging

Atomic force microscopy was performed using a NanoWizard AFM system (JPK Instruments, Berlin, Germany). The AFM head was mounted on top of a stable cast-iron microscope stage and combined with a LSM 510 confocal microscope (Carl Zeiss, Jena, Germany). Soft, rectangular silicon cantilevers (CSC38/noAl, Micromash, Tallin, Estonia) with a nominal spring constant of 0.03 N/m were used. The cantilever sensitivity in V/m was determined before each measurement. The spring constant was calibrated by using the thermal fluctuations method. Clean, circular glass cover slips (d = 24 mm, #1.5, Menzel Gläser, Thermo Fisher, Braunschweig, Germany) were hydrophilized by air plasma cleaning. The AFM fluid cell was assembled by using the glass slide and filled with 400 µl of the reaction buffer. Directly before AFM imaging, myofilaments were diluted with reaction buffer to a concentration of 10 nM. 5 µL of 10 nM myofilaments in reaction buffer were added to the fluid cell. In experiments with combined AFM and fluorescence imaging, Alexa-488 labeled myofilaments were used. After 15 min incubation most of the myofilaments adhered to the hydrophilic surface, residual non-adherent filaments were removed by washing with reaction buffer. AFM imaging was performed in contact mode with a scan rate of 1 Hz. The imaging forces were kept very low (<0.5 nN) by continuously adjusting the deflection setpoint and using optimized feedback gains. Raw AFM images were line and plane fitted by using the open source software Gwyddion (www.gwyddion.net).

## Myofilament length determination

Length histograms of myofilaments were obtained from confocal images of Alexa-488 labeled myofilaments prepared as described in the AFM imaging section. Confocal microscopy was performed using a LSM 510 Meta system (Carl Zeiss, Jena, Germany) with a 40× water immersion objective (C-Apochromat, 40×/1.2, Carl Zeiss). The 488 nm line of an Argon-ion laser was used to excite the sample. Fluorescence excitation and emission were separated using a microscope build-in dichroic mirror (HFT 488/633) and band pass filter (BP 505–550) in front of the detector. Myofilament length measurement of the AFM images was manually performed with the software Gwyddion. Myofilament length distribution of the confocal images with xy-pixel size of 110 nm was obtained manually using Image J. The accuracy of the length determination based on confocal microscopy was verified by colocalization with AFM images. Identical lengths of the filaments were obtained using both methods (Pearson's $r$ = 0.99, n = 47) and therefore justified bulk measurements using confocal microscopy.

## Data analysis

Data analysis was performed with Image J (Rasband, W.S., National Institutes of Health, USA, http://imagej.nih.gov/ij) and custom written scripts in Igor Pro 6.0 (WaveMetrics, Lake Oswego, USA)

and MatLab. Actin filament length measurement (*Figure 2B*) was performed using the NeuronJ Image J plugin, for details see (*Meijering et al., 2004*). The fluorescence intensity profiles for *Figure 2D,E* (main text) were obtained using the segmented line tool and plot profile command in Image J. To determine the fluorescence intensity in the area occupied by a myofilament (*Figure 3C* [main text]) a custom written macro in Image J was used. In brief, the image sequence in the 647 nm channel of the Alexa-647 labeled myofilament (red) was converted to 8-bit, binarized by using a threshold filter and an area selection (corresponding to the myofilament) for each binary image was created. The mean fluorescence intensity in the corresponding 488 nm channel images of the Alexa-488-phalloidin labeled actin filaments (green) was computed in the selected area. Smoothing of intensity profiles in *Figure 3C* was conducted using a sliding-average smoothing algorithm (interval 2 s) implemented in Igor Pro.

Myofilaments for the velocity analysis were tracked using a custom written program by Rogers et al. (for details see (*Rogers et al., 2007*)). The radial velocity $v_t$ was computed from the myofilament trajectories $x_t$, $y_t$, where t is the discrete sampling time with an interval $\Delta t = 0.2$ s. The radial change in position was computed by $\Delta r_t = ((x_{t+n\Delta t} - x_t)^2 + (y_{t+n\Delta t} - y_t)^2)^{1/2}$. The radial velocity was obtained from $v_t = \Delta r_t/n\Delta t$. Computation intervals $n\Delta t$ with n = 20 were chosen to reduce the noise induced by small fluctuations in the positions.

## Acknowledgements

We thank J Käs and the Käs lab for providing us the myosin purification protocol and J Howard for discussion and comments on the manuscript.

## Additional information

### Funding

| Funder | Grant reference number | Author |
|---|---|---|
| Gottfried Wilhelm Leibniz-Program of the DFG | SCHW716/8-1 | Sven Kenjiro Vogel, Petra Schwille |
| Max Planck Society | | Sven Kenjiro Vogel, Zdenek Petrasek, Fabian Heinemann, Petra Schwille |
| Daimler und Benz foundation | 32-09/11 | Sven Kenjiro Vogel, Fabian Heinemann |

The funders had no role in study design, data collection and interpretation, or the decision to submit the work for publication.

### Author contributions

SKV, Conception and design, Acquisition of data, Analysis and interpretation of data, Drafting or revising the article; ZP, developed the theory and performed the simulations, Analysis and interpretation of data, Drafting or revising the article; FH, acquired and analyzed the AFM data and helped with the analysis of the experimental data, Acquisition of data, Analysis and interpretation of data, Drafting or revising the article; PS, Drafting or revising the article.

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
