## [Author Response]

*1. There is concern at the lack of quantification of the major experiment. There is a need to quantify these results. How many times were these experiments performed and how many times were the effects of myofilaments on actin filaments observed*?

We agree with the comments of the referees and we have added a detailed quantification of the processes within the main text. The description of all experiments in the text was modified accordingly.

Quantitative data was added as follows: “… Actin pattern formation occurred at ATP concentrations between 0.1 - 1 µM in systems where ATP is enzymatically regenerated (Table S1) in 94 % of the experiments (n = 45 experiments).”

The following sentence was added: “…Strikingly, myofilament addition to low density MACs displayed breakage events and compaction of actin filaments, resulting in their shortening over time in all experiments (n = 21, Figure 2 A–C, Movie S2).”

And the following sentence was added: “…68 % of the individually observed myofilaments displayed directed movement and 50 % exhibited fragmentation and compaction of an actin filament (total number of myofilaments = 152; 7 experiments). In cases where both fragmentation and compaction occurred, 95% of the observed myofilaments showed directed movement along the actin filaments, while 5 % remained stationary at their original binding site (total number of fragmenting myofilaments = 75; 7 experiments).”

*2. It is not clear how much the simulations have helped the paper. It would be more useful if the simulations lead to some testable hypothesis that then expand/strengthen/fine-tune the model. This needs to be clarified*.

The simulations are not supposed to lead to a new hypothesis, but rather (in first iteration) show that the already suggested model of interaction between the actin and myosin filaments is plausible, given the quantitative parameter values known from literature (rate constants). Most importantly, it shows that sufficient force needed to bend and break the actin filament can be generated by a single myofilament, assuming only the difference in the interaction between the leading and trailing myofilament ends (making a step vs. not making a step). This conclusion does not follow simply from the qualitative formulation of the model, as shown in Figure 3D.

In addition, the simulation yields quantitative details of the model, for example, the mean fractions of myosin heads in the 6 possible states (now included in the new Figure 9). The simulations also describe the fluctuations of the actin filament curvature (Figure 4C, D).

Thus, the simulations are an essential part of this work, as they provide quantitative support for our qualitative model (Figure 3D) and give detailed insight into the molecular mechanism.

To clarify this point, we added Figures 8 and 9. In Figure 9, we show the fractions of the myosin heads in the different states of the motor cycle illustrated in Figure 8, and we have included a new paragraph in the discussion.

*3. Most experiments (at least those employed to derive mechanism) were performed in near rigor conditions. Given that the km for myosin-II nucleotide-binding is in the tens of micro-molar, the use of ∼3 micro-molar ATP is a concern. With respect to muscle (the source of the myosin-II in these experiments), the ATP concentration is ∼5 mM and >100 micro-molar in non-muscle cells*.

To bend and break the actin filament, the myofilament has to be attached to the actin for sufficient time, and sufficient force has to be generated, requiring a certain mean number of heads being attached to actin at any given time. Given the small number of interacting myosin heads, this translates into a requirement that a large fraction of heads is in the bound state. In our assay, this was achieved at low ATP concentrations. Here, it is important to mention that skeletal muscle myosin has a lower duty ratio and is therefore less processive than non-muscle myosin in cells (Harris De at. al 1993; Wang et. al. 2003). By lowering the ATP concentration in our assay, we increased the duty ratio of our myofilaments and thereby made them more processive (Figure 5, Movie 5), similar to non-muscle myosin, which is in line with previous studies (Soares et. al. 2011; Smith et. al. 2007; Humphrey et. al. 2002).

Moreover, the modification of the kinetic rates of the motor cycle could be controlled *in vivo* through phosphorylation of the myosin light chains (Tan et. al. 1992; DeBiasio et. al. 1996; Matsumura et. al. 2001) and possibly allows the behavior observed *in vitro* to occur at the higher ATP concentrations found *in vivo*. We would like to emphasize that the aim of this study was to give insight into the individual myosin-actin interactions by using a minimal system. It is obvious that due to the simplicity of the system, parameters such as the ATP concentration may differ from the *in vivo* values. To clarify this point we extended the Discussion.

*The authors predict that 30 molecules of myosin-II (per mini-filament) are in contact with an actin filament. However, the force estimate (23 pN) for buckling only reflects that of 5–6 active heads. In other words, >80% of the myosin heads are in rigor*.

In our simulations, all myosin heads are active and constantly move through the six states of the motor cycle (now added for clarity of presentation as new Figure 8). Changing the ATP concentration will change the average residence time of the individual heads in the six states of the motor cycle. As stated by the reviewers, a low ATP concentration will increase the number of heads in the rigor state (corresponding to state 5 in Figure 8). However, this state is a part of the biochemical motor cycle, and a higher population of heads in state 5 does not indicate that these heads are ‘inactive’, but describes the fact that individual filaments populate the fifth (rigor) state longer (in temporal average) before proceeding to the next state. Hence, all heads are active and only the distribution of states in the motor cycle changes. To further illustrate this aspect, the mean fractions of heads in the individual states are now shown in the new Figure 9.

*This type of tension/buckling is what is expected (and previously seen) in traditional motility assays when ATP is kept low. This point assumes the authors are using fully active myosin-II preparations (based on their testing of preparations)*.

The referees may refer to a study from Kron and Spudich (1986) where actin filaments move on synthetic myofilaments that were fixed to a glass surface. In this study, it was observed that some filaments broke into fragments after they bound to the myosin-coated surface. In contrast to our assay, in these experiments filaments broke at high (1 mM) ATP concentrations, and condensation into foci was not observed. Moreover, the surface density of bound myofilaments had to be high, as actin filaments break less frequently on a low-density myosin substratum. This suggests that breakage of actin filaments in their study occurred because of interaction with at least two or multiple myofilaments. Last, their myofilaments are fixed to a support, while our myofilaments are not tethered to a support.

In our study, we directly show that one myofilament is capable to fragment and compact actin filaments while it is directly moving along the actin filament. Fragmentation here is based on compressive stress generated within one myofilament, while in the motility assay it can not be distinguished whether pulling forces or compressive stresses play the dominant role, and how many myofilaments are involved.

*4. The authors use myosin-II mini-filaments that are free in solution, which forces them to use rigor conditions. Given what the authors are aiming to reproduce, shouldn't myosin-II be tethered independently of actin in their assays (as it is presumably to some degree at the cell division site, given that myosin localization is often tail-dependent and actin-independent)? This approach would invoke tension (with actin and myosin tethered) reflecting the force-velocity relationship of myosin-II, an effect that is more likely seen in cells*.

Myofilaments also bind to actin filaments at high (non-rigor) ATP concentrations (Movie 5, Figure 5). Only the time that myofilaments stay attached to the actin filaments is much shorter due to the low processivity at higher ATP concentrations.

The aim in this work was not to reproduce exactly the situation in a cell (if this is ever applicable), but to study the actin–myofilament interaction in a simple, well-defined system, and to see how the observed behavior could be explained, and what consequences this could have for the *in vivo* situation. We directly show that single myofilaments can interact with actin in such a way that sufficient force can be generated to break the filament, without either the myofilament or the actin being firmly attached to a solid support or scaffold. Note that membrane-anchored actin exhibits lateral diffusion on the membrane. *In vivo* myosin localization may indeed depend on certain domains of the myosin protein but does not necessarily imply that myosin is physically fixed to the cell membrane or any rigid scaffold. It is more likely that myosin interacts *in vivo* with other cortical proteins that help to maintain their cortical (equatorial) localization in cells (Piekny et al., 2008). Typically protein–protein interactions are transient and can be best described by attachment and detachment rates. Therefore fixation of myosin to a support in our *in vitro* assay would in our opinion represent a rather non-physiological situation.

*5. How closely does the actin and myosin concentration used reflect the in vivo actomyosin density at the cell cortex*?

Cortical actin density varies in cells. Known median mesh diameters range from ∼ 40 to 230 nm (Morone et al., 2006). We can control actin density such that single actin filaments are visible or that the density of the actin mesh is so high that individual actin filaments can not be distinguished by TIRF microscopy, due to the diffraction limit of light (see Figure 1B). This density may be comparable to the *in vivo* situation and we also observed actin pattern formation at higher actin densities. Nevertheless the prior aim was to investigate the interaction of individual myofilaments with single actin filaments, which required the use of a lower actin density for the main experiments.

*The choice of muscle myosin-II seems specialized; perhaps non-muscle myosin-II would be best for reconstituting general cellular behavior*?

See response to comment 6, below.

*6. We generally felt that the conditions used in the experiments were not very physiological and hard to reconcile with what is probably found in cells. Two types are experiments need to be performed to further validate the conclusions: (a) perform the study with non-muscle myosin II at physiological ATP concentrations and (b) attempt to link myosin II, rather than F-actin, to the minimal cortex to observe effects on F-actin shortening, buckling, and turnover*.

Muscle myosin II is a standard tool to mimic actomyosin contractility present in cells (Backouche, Haviv, Groswasser, & Bernheim-Groswasser, 2006; Kohler, Schaller, & Bausch, 2011; Smith et al., 2007; Soares e Silva et al., 2011). Our results now can be directly compared to these studies.

Our assay represents a specific minimal system to allow elucidating fundamental mechanisms of the actin myosin interaction. We strongly believe that the potential new insights by using non-muscle myosin II (i.e., redoing exactly the same work as reported, albeit with another protein system) would be limited, since also in this case, a minimal system is used and other interacting proteins are absent. It is therefore highly likely that the ATP dependency would also in this case deviate from physiological conditions.

Moreover, linking myosin II to a support is rather non-physiological, as discussed in greater detail in the answer to comment 4.

Besides these concerns, we agree with the reviewer that using our assay with non-muscle myosin II to compare the two proteins is definitely worth investigating and we will perform such studies in the future.